# Land-Cover Classification of Coastal Wetlands Using the RF Algorithm for Worldview-2 and Landsat 8 Images

**Xiaoxue Wang [1], Xiangwei Gao [1],\*, Yuanzhi Zhang [2,3], Xianyun Fei [1], Zhou Chen [1], Jian Wang [1], Yayi Zhang [1], Xia Lu [1] and Huimin Zhao [1]**

1   School of Geomatics and Marine Information, Jiangsu Ocean University, Lianyungang 222002, China
2   School of Astronomy and Space Science, University of Chinese Academy of Sciences, Beijing 100049, China
3   National Astronomical Observatories, Key Lab of Lunar Science and Deep-Space Exploration, Chinese Academy of Science, Beijing 100101, China
\*   Correspondence: gaoxw@hhit.edu.cn

**Abstract:** Wetlands are one of the world's most important ecosystems, playing an important role in regulating climate and protecting the environment. However, human activities have changed the land cover of wetlands, leading to direct destruction of the environment. If wetlands are to be protected, their land cover must be classified and changes to it monitored using remote sensing technology. The random forest (RF) machine learning algorithm, which offers clear advantages (e.g., processing feature data without feature selection and preferable classification result) for high spatial image classification, has been used in many study areas. In this research, to verify the effectiveness of this algorithm for remote sensing image classification of coastal wetlands, two types of spatial resolution images of the Linhong Estuary wetland in Lianyungang—Worldview-2 and Landsat-8 images—were used for land cover classification using the RF method. To demonstrate the preferable classification accuracy of the RF algorithm, the support vector machine (SVM) and *k*-nearest neighbor (k-NN) methods were also used to classify the same area of land cover for comparison with the results of RF classification. The study results showed that (1) the overall accuracy of the RF method reached 91.86%, higher than the SVM and k-NN methods by 4.68% and 4.72%, respectively, for Worldview-2 images; (2) at the same time, the classification accuracies of RF, SVM, and k-NN were 86.61%, 79.96%, and 77.23%, respectively, for Landsat-8 images; (3) for some land cover types having only a small number of samples, the RF algorithm also achieved better classification results using Worldview-2 and Landsat-8 images, and (4) the addition texture features could improve the classification accuracy of the RF method when using Worldview-2 images. Research indicated that high-resolution remote sensing images are more suitable for small-scale land cover classification image and that the RF algorithm can provide better classification accuracy and is more suitable for coastal wetland classification than the SVM and k-NN algorithms are.

**Keywords:** coastal wetland; classification; RF algorithm

## 1. Introduction

Wetlands play an important role in every aspect of ecological systems, including by providing habitats for fish and wildlife [1,2], regulating climate, improving water quality [3], sequestering carbon [4], and mitigating flooding [5]. However, most wetlands, especially coastal wetlands, on the one hand, are disappearing or are in danger of disappearing as a result of rapid economic development and population growth that lead to urban sprawl. On the other hand, the rising sea

level caused by global climate change has also led to a rapid decline in the area of coastal wetlands. Accordingly, governments must plan, manage, and reconstruct coastal wetlands through wetland protection activities that will require the ability to quickly, precisely, and repeatedly obtain information about coastal wetland land cover for large areas [6]. Remote sensing (RS) technology has become one of most important methods for acquiring such information, and much effort has been given to improving various RS classification methods so as to obtain more valuable land cover classification results [7,8].

Classification of coastal wetlands is challenging, not least because of their complicated composition and pattern [9]. In previous studies, Landsat MSS, Landsat TM, and SPOT images have predominantly formed the basis of remote sensing techniques used for wetland monitoring and for land cover mapping, extraction, and classification [10–13].

Owing to technical limitations, the resolution of such images is relatively low, making them suitable only for large-scale area research and unable to provide detailed land use information. Aerial photographs are a very efficient method for acquiring micro-scale wetland cover mapping. At present, however, professional aerial drone operators are needed to take aerial photographs, most existing aerial photographs already meet commercial needs, and aerial photographs for research can be difficult to obtain.

Compared with aerial photographs, high-resolution satellite remote sensing images have the advantages of saving manpower and material resources, being easily obtainable, being less time-consuming, and offering higher accuracy. Recently, high-resolution remote sensing images at the meter and sub-meter levels, such as IKONOS (1 m), Quick Bird (0.61 m), and Worldview-2 (0.5 m), have been widely used in various fields, including land use classification, urban planning, and environmental monitoring [14–20]. For high spatial resolution remote sensing images, the spatial structure was fully expressed and the interior spectral valve was more changeable for each land cover type, increasing the heterogeneity of images, so that traditional pixel-based classification technologies are not suitable. The object-oriented method, however, provides technical support for high spatial remote sensing wetland land cover classification [21]. Using this method, pixels are merged into an entity unit—that is, the object. Merged objects exhibit a great many features, including textures, shapes, locations, relative relationships, and distances, as well as other non-spectral features. However, because massive object features pose great challenges to classification, the choice of the classification algorithm is essential for making full use of object characteristics.

The selection of classification algorithms is a key issue in classification, greatly affecting as it does the accuracy of land cover classification. Traditional classification methods include the K-mean, the minimum distance method, the maximum likelihood method, the *k*-nearest neighbor (k-NN) method, and the like; advanced algorithms include the decision tree, artificial neural network, support vector machine (SVM), and random forest (RF) methods. These classification algorithms were applied to various images for land use classification of different surface conditions and types of patterns. Neural networks, decision trees, and SVM are widely used in land cover classification, soil type classification, vegetation (urban vegetation, tree species, and crops) classification, water quality detection, and the like [22–28]. In wetland classification and land cover studies, researchers make their own selections based on their needs. Sader et al. [12], for example, used Landsat-TM images to conduct experiments in Orono and Acadia and proposed a combination of mixed classification and GIS-based methods as the most promising method for classification of forest wetlands. Augusteijn et al. [29] used optimized radar data and neural networks to distinguish between wetlands and uplands. However, although these classifications achieved excellent results, overall accuracy remained below 85%.

The algorithms introduced and used in this article are k-NN, SVM, and RF. The k-NN classification algorithm, one of the simplest machine learning algorithms, is a theoretically mature method [30,31]. The principle of the k-NN algorithm is that if most of the k nearest samples of a sample in the feature space belonged to a certain category, the sample is also divided into this category. In this method, the classification of the samples to be divided is determined only according to the category of one or more nearest samples. However, its main disadvantage is that when the sample is unbalanced

and a new sample is input, one of the K neighbors of the sample is the majority, which is easy to cause misclassification. Another disadvantage of this method is that it involves a large number of calculations: The classification of each sample needs to calculate its distance from all known samples in order to obtain its K nearest neighbors [32]. SVM is a supervised learning model commonly used for pattern recognition, classification, and regression analysis. Nonlinear mapping forms the theoretical basis of SVM, whose goal is the optimal hyperplane of feature space partition. SVM is a novel small sample learning method with a solid theoretical basis. For classification of land cover types, however, it has major disadvantages. For example, solving the multi-class classification problem is difficult with SVM. The classical SVM algorithm gives only two classes of classification algorithm, but in the practical application of data mining, it is generally necessary to solve the problem of multi-class classification [33]. Notably, Salehi et al. [34] and Lin et al. [35] have proposed an improved SVM algorithm that uses different methods and with which they obtained excellent overall accuracy and kappa coefficient. Salehi et al. [34] compared the improved result with the accuracy of the MLC classification and found the improved SVM method to be superior.

Based on bootstrap sampling, the RF method is an extended variant of bagging. This efficient integrated learning algorithm can be used for multi-class classification, regression, and other tasks without modification. The advantage of the RF method is that the algorithm is simple, is easy to implement, requires little calculation time, is suitable for a large number of characteristic parameters, is effective at solving redundancy problems, and exhibits a powerful performance in many practical applications. These advantages can make up for the shortcomings of k-NN and SVM while obtaining better classification results [36]. Since its proposal in 2001, this method has been widely used in ecology and land use classification. This highly precise algorithm allows determination of variables' importance, prediction of complex interactions between variables, and reduction of training samples. Mahdianpari et al. [37] used SAR data to classify wetlands in the northeastern portion of the Avalon Peninsula and obtained an overall classification accuracy of 94% using the RF algorithm. In arid areas of Xinjiang, Tian et al. [38] used the RF algorithm to classify the wetlands, obtaining an overall accuracy of 93%. Using RF and other methods in combination can produce still better results. Liu et al. [39] combined RF and multiple end-member spectral mixture analysis for wetland classification in the Yellow River Delta area using Landsat-8 images. The experiments showed the RF algorithm to be suitable for the classification of wetlands in the Yellow River Delta area and revealed that the combination with multiple end-member spectral mixture analysis could improve results by 2%–3%. At present, the RF algorithm is notably popular, as research has shown that it performs well in a variety of complex wetland classifications. For classification of estuary wetland land cover, however, the RF method has seen less application for high-resolution remote sensing images, offering a new direction for research.

This paper compared the application of Worldview-2 image and Landsat-8 image in the classification of land cover types in coastal wetlands in small areas. The classification accuracy of RF, SVM, and k-NN algorithms in coastal wetland land cover were compared respectively in Worldview-2 image and Landsat-8 image. Texture features were used for classification to further improve the classification accuracy.

## 2. Data and Method

In this paper, two types of remote sensing images, Worldview-2 high-resolution images and Landsat-8 remote sensing images, of the Linhong Estuary region in Lianyungang City were used to classify the study area using the RF classification method. The classification results obtained by use of the SVM and nearest neighbor methods were compared with the results of the RF classification. All classifications were performed based on an object-oriented segmenting process in which a large number of characteristics, including shape features, texture characteristics, and spectral information, were used to classify the image. The high resolution remote sensing image distinguished well between objects having similar spectral features and improved the classification accuracy of these

objects. Classification features included the additions of texture features that allowed the comparison with classification results for shape features and spectral features. To highlight the superiority of Worldview-2 high-resolution images for the classification of wetland land cover, Landsat-8 remote sensing imagery of the region was used for land cover analysis, with the comparison of the classification accuracy of the RF, SVM, and k-NN methods for various images. The parameter settings of the three classification methods were the same in the two images. In the RF method, a random forest consisting of 100 decision trees was established. For each decision tree, the maximum depth was set to 100. In the SVM method, the radial basis function was used to determine the hyperplane, and the c parameter was set to 2. In the k-NN method, the k parameter was set to 2.

## 2.1. Study Area

The study area was Linhong River Estuary Wetland, in the northeast of Lianyungang City, with a longitude/latitude of 119°11′37″–119°16′33″ E and 34°45′37″–34°49′15″ N (Figure 1) and an average elevation of 3–5 m. The total area of the experimental image was 27.6 km$^2$. Linhong River Estuary is the mouth of Xinshu River and is the sea channel of regional rivers such as the Rose River, Fan River, and Zhuji River. Covering tens of square kilometers, it plays a role in flood discharge and helps drain and tail water into the sea in Lianyungang. It is located in a warm temperate humid monsoon climate and has an average precipitation of 900.9 mm [40–43]. A large number of artificial and natural objects, including salt field, fish pond, and tidal flat types, are distributed on either side of the river. In 2001, the Linhong River Estuary in the coastal zone of Lianyungang City was used for sewage discharge and flood discharge [44]. In recent years, the overall level of pollution in the sea area of Lianyungang City has been relatively high, and ineffectively controlled discharges of land-source pollutants have seriously polluted the Linhong River Estuary. Simultaneously, reclamation efforts have decreased coastline resources as the natural coast has been cut and straightened, the natural coastline reduced, and the natural coastal landscape and offshore ecological environment destroyed [45]. However, the latest Lianyungang marine economic development plan has named the Linhong River Estuary as a key protected area, and a new Linhong River wetland reserve will be built to restore the wetland ecological environment as well as the area's biodiversity [46].

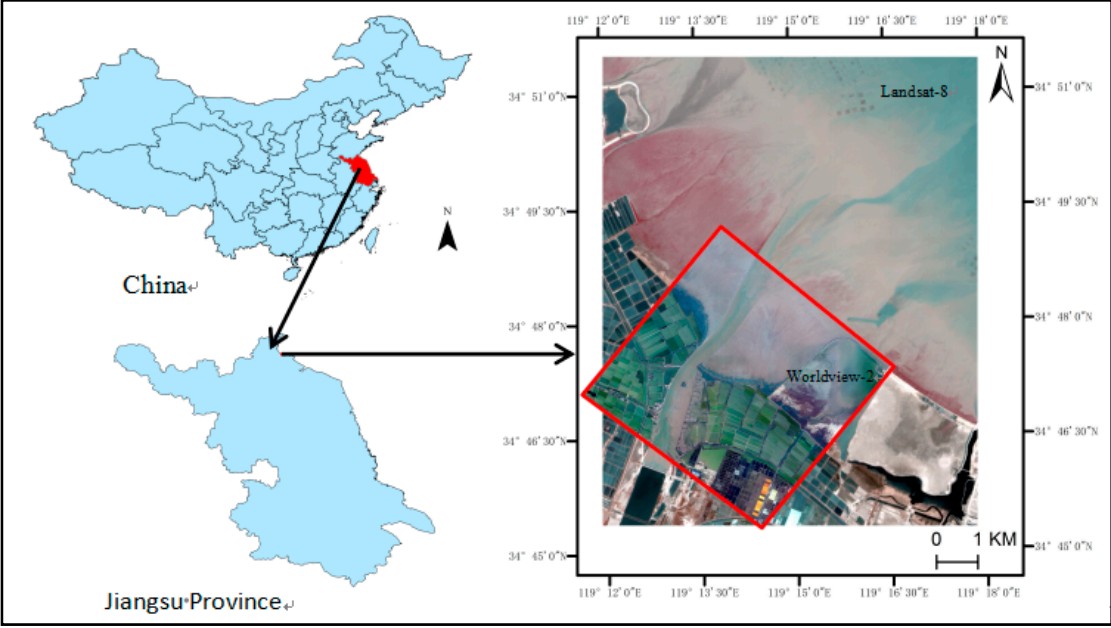

**Figure 1.** Study area.

## 2.2. Satellite Data

Two different spatial resolution images were used for classification in this study area (Table 1). Worldview-2 satellite imagery consisted of four standard bands (blue, green, red, and NearIR-1), four new bands (coastal, yellow, red-edge, and NearIR-2) at a spatial resolution of 2 m, and a panchromatic band at a spatial resolution of 0.5 m. Of the four newly added bands, the coastal band is greatly influenced by the atmosphere and can be used to improve atmospheric correction technology; the yellow band can detect yellow vegetation on land and water, important for feature classification; the red-edge band, located between the red band and the near infrared band, is useful for classification of vegetation; and the NIR-2 band coincides with the NIR-1 band but is less affected by the atmosphere [47,48]. The image acquisition used in the experiment was made on 1 October 2012, and image fusion was performed prior to the experiment using the Gram-Schmidt Pan Sharpening method [49].

**Table 1.** Band Descriptions.

| Band | Worldview-2 | | Landsat-8 | |
|------|-----------------|-----------------|-----------------|-----------------|
| | Wavelength (nm) | Spectral Region | Wavelength (nm) | Spectral Region |
| 1 | 400–450 | Coastal | 433–453 | Coastal |
| 2 | 450–510 | Blue | 450–515 | Blue |
| 3 | 510–580 | Green | 525–600 | Green |
| 4 | 585–625 | Yellow | 630–680 | Red |
| 5 | 630–690 | Red | 845–885 | NIR |
| 6 | 705–745 | Red edge | 1560–1651 | SWIR1 |
| 7 | 770–895 | NIR-1 | 2100–2300 | SWIR2 |
| 8 | 860–1040 | NIR-2 | 500–680 | Pan |

The Landsat-8 image used for the experiment was taken in April 2013, and the image was fused using the Gram-Schmidt Pan Sharpening method to obtain a multispectral image with a resolution of 15 m.

## 2.3. Training and Verification Data

Due to wetland protection and terrain restrictions, the seaside area could not be reached. It could only be investigated along narrow roads. The location of the surveying was recorded using the GNSS system. Two months of field surveys were conducted to obtain land cover in the study area, establishing a mapping between actual land use types and remote sensing images. Multi-scale segmentation was carried out in eCognition [50], and the image was automatically segmented into different polygons. The training samples and the verification samples were randomly selected in these polygons by visual interpretation and evenly distributed over the entire image. Random selection of the samples within each category assured that the samples were representative for each category. Due to the different image resolutions, the level of detail of land cover types was different. Therefore, different classification systems should be established, and different samples should be selected according to the classification system. The land cover types seen on Worldview-2 images were tidal flat, vegetation, building, backfill land, estuary, fish pond, road, saline alkali soil, and salt land. The land cover types seen on Landsat-8 images were tidal flat, road, vegetation, building, aquaculture area, backfill land, estuary, fish pond, and high-salinity area (Tables 2 and 3, Figure 2).

**Table 2.** Class description training samples and verification samples in the Worldview-2 images.

| Class | Worldview-2 Image | Class Description | Training Samples | Verification Samples | Actual Average Area of the Samples (m$^2$) |
|---|---|---|---|---|---|
| Tidal flat | | Close to the river estuary Larger area Image smoothing | 78 | 67 | 5985 |
| Vegetation | | Displayed in green under the true color band combination, can be extracted with NDVI | 78 | 49 | 1391 |
| Building | | Regular arrangement Centralized distribution away from the estuary | 30 | 36 | 157 |
| Backfill land | | Artificial backfill dark brown in true color band. | 19 | 21 | 3218 |
| Estuary | | Gradually widened by a narrow strip, can be extracted with NDVI | 57 | 47 | 4055 |
| Fish pond | | Blue square or rectangle, it can be extracted with NDWI | 51 | 55 | 7896 |
| Road | | Slim shape, distributed along the building area | 9 | 14 | 1629 |
| Saline alkali soil | | Irregular shape, high brightness | 30 | 16 | 1283 |
| Salt field | | Available in a variety of colors, rectangular in shape | 9 | 9 | 3119 |

**Table 3.** Class description training samples and verification samples in the Landsat-8 images.

| Class | Landsat-8 Image | Class Description | Training Samples | Verification Samples | Actual Average Area of the Samples (m$^2$) |
|---|---|---|---|---|---|
| Tidal flat | | Close to the river estuary Larger area image smoothing | 30 | 32 | 120,006 |
| Vegetation | | Displayed in green under the true color band combination, can be extracted with NDVI | 17 | 15 | 84,642 |
| Backfill land | | Artificial backfill dark brown in true color band. | 9 | 10 | 39,889 |
| Estuary | | Gradually widened by a narrow strip, can be extracted with NDVI | 15 | 17 | 169,450 |
| Fish pond | | Blue square or rectangle, it can be extracted with NDWI | 35 | 44 | 33,686 |
| Road | | Slim shape, distributed along the building area | 20 | 22 | 34,837 |
| High salinity area | | Irregular shape, high brightness | 21 | 20 | 10,361 |
| Aquaculture area | | Distributed in the sea, dark squares or rectangles | 9 | 9 | 9575 |

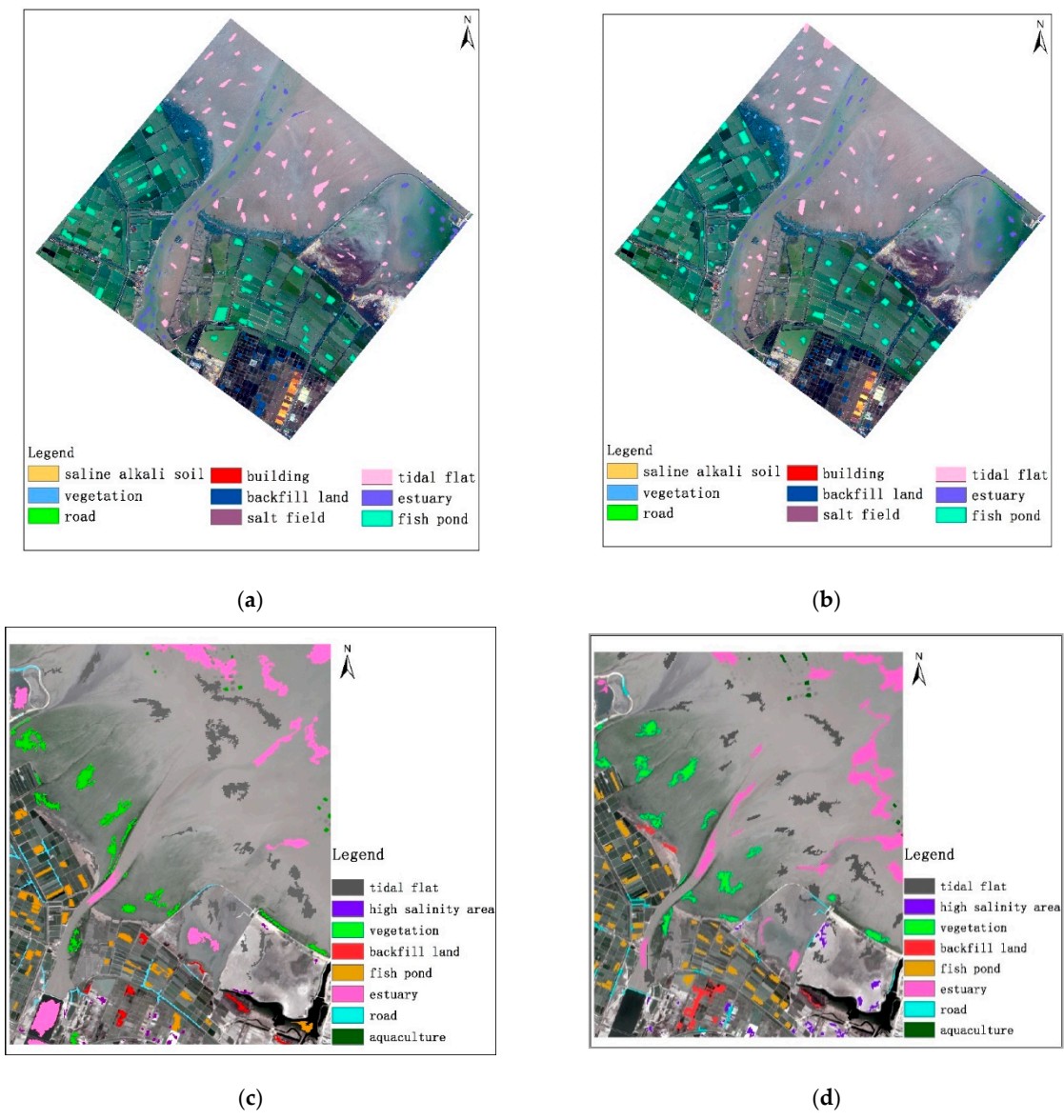

(**a**)　　　　　　　　　　　　　　　　　　(**b**)

(**c**)　　　　　　　　　　　　　　　　　　(**d**)

**Figure 2.** Worldview-2 and landsat-8 training sample and verification sample distribution. (**a**) Worldview-2 training sample; (**b**) Worldview-2 verification sample; (**c**) Landsat-8 training sample; and (**d**) Landsat-8 verification sample.

*2.4. Segmentation*

The experiment began with a determination of the segmentation scale, in the image, each segmented object corresponded to the actual feature as much as possible. Specific object types could be expressed by one or several objects. The shape of the object boundary should not be overly broken, the spectral differences of objects in the same class should be small, and the boundaries of features should not be blurred [51]. Image segmentation and subsequent selection of experimental features, classification, and assessment of accuracy were completed in eCognition. Several key parameters were set during multi-scale segmentation, including shape, compactness, and scale. The sum of shape and color is 1, with shape determining the proportion of image spectral (color) information and object shape information in the segmentation process. If the shape is 0.3, for example, then color is 0.7. The sum of smoothness and compactness is also 1, with compactness controlling how much the object's shape tends to be spatially compact versus spectrally homogeneous (smooth) but less compact. The scale parameter limits the color and shape complexity of the overall object [52]. ESP [53],

a tool proposed by Dragut for automatically obtaining the optimal segmentation scale parameters for the multi-scale segmentation algorithm in eCognition, was used as a scale evaluation tool for this experiment. The optimal segmentation scale parameters were determined from the evaluation results of ESP tools, visual discrimination of the segmentation effect, and the actual situation of surface features. The results of the ESP calculation were obtained, and three representative peaks, 126, 86, and 53, were selected for segmentation experiments (Figure 3).

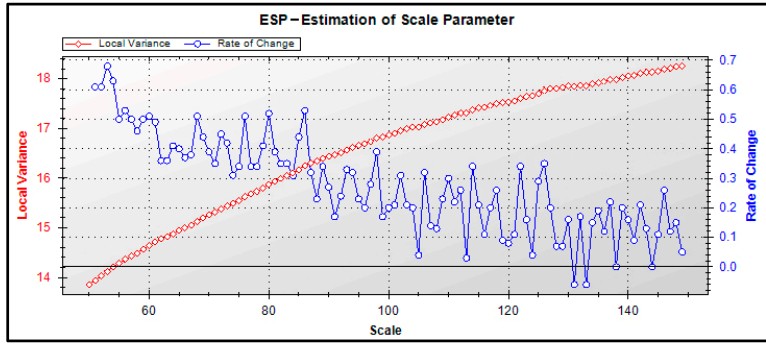

**Figure 3.** The result of the ESP calculation.

As shown in Figure 4, at a segmentation scale of 126, the edge of the building could not be represented, but at a scale of 53, the complete building was too fragmented—rather, it was better reflected at a scale of 86. The experimental results show that the combination of parameters at which the shape was 0.2, compactness 0.5, and segmentation scale 86 could be applied for segmentation of the image. In the Landsat-8 image, by contrast, the segmentation scale was 120, the shape 0.1, and the compactness 0.5.

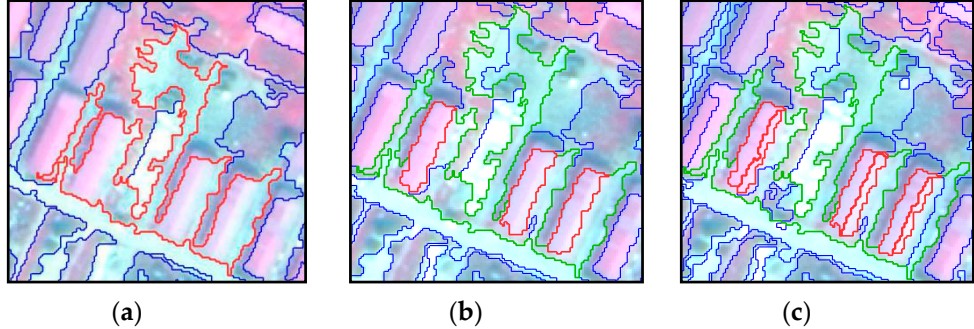

| (a) | (b) | (c) |

**Figure 4.** The segmentation effect at different scales. (**a**) Shape 0.2, compactness 0.5, and segmentation scale 126; (**b**) shape 0.2, compactness 0.5, and segmentation scale 86; and (**c**) shape 0.2, compactness 0.5, and segmentation scale 53.

## 2.5. Features

Spectral features, index features, geometric features, and texture features were used for land cover classification of coastal wetlands. In each image, 80 features were included in the classification: Nine spectral features, two index features, five geometric features, and 64 texture features. The spectral features included the mean of each band and brightness. The index features included normalized difference vegetation index (NDVI) and normalized difference water index (NDWI). The geometric features included asymmetry, boundary index, compactness, density, and shape index (Table 4).

**Table 4.** Partial feature description.

| Feature | Description |
|---------|-------------|
| NDVI | $\frac{p(NIR)-p(R)}{p(NIR)+p(R)}$ |
| NDWI | $\frac{p(G)-p(NIR)}{p(G)+p(NIR)}$ |
| Asymmetry | Expressed as the length ratio of the short axis and the long axis of the ellipse. |
| Border index | Represented the ratio of the true circumference of the object to the circumference of the smallest enclosed rectangle of the object. |
| Compactness | Describe the compactness of an image object, which is the product of length and width divided by the number of pixels. |
| Density | Describe the degree of compactness of the object. The closer the object is to the square, the higher the density is. |
| Shape index | The length of the boundary of the image object divided by four times its square root of the area. The more the objects are broken, the larger the shape index is. |

The gray level co-occurrence matrix method was used to extract texture features and obtain the features of homogeneity, contrast, dissimilarity, entropy, ang. 2nd moment, mean, std. dev., and correlation [54]. Eight different texture features were calculated for eight bands of the WV-2 image, totaling 64 texture features (Table 5).

**Table 5.** Texture feature description.

| Feature | Description |
|---------|-------------|
| Homogeneity | It is a measure of the local gray uniformity of the image. The larger the numerical value, the more uniform the texture of the image is. |
| Contrast | It reflects the clarity of the image and the depth of the texture grooves. The deeper the texture grooves, the greater the contrast and the clearer the visual effect. On the contrary, if the contrast is small, the groove is shallow and the effect is blurred. |
| Dissimilarity | It is similar to contrast, but increasing linearly. The value increases with the increasing of local contrast. |
| Entropy | It is the clarity of the image, which is the degree of texture clarity. |
| Ang. 2nd moment | It is a measure of the uniformity of gray distribution of the image. The texture is thicker, and the value is larger. |
| Mean | It reflects the regularity of texture. When the texture is cluttered and difficult to describe, the value is small. When texture is regular and easy to describe, the value is larger. |
| Std. dev. | It is a measure of the deviation between the pixel value and the mean value. When the gray level of the image changes greatly, the value is large. |
| Correlation | It reflects the local gray correlation in the image. When the values of matrix elements are uniform and equal, the correlation values are large. On the contrary, if the difference of matrix pixel values is large, the correlation value is small. |

*2.6. Accuracy Evaluation*

There are many methods of evaluating classification accuracy, including the confusion matrix and ROC curve. The former is commonly used in land cover classification: The column of the matrix is the reference image information, the row is the information of the evaluated image classification result, and the portion where the row and the column intersect gives the number of samples classified into a specific category related to the reference category.

The main indicators are the producer (PA), user (UA), overall accuracy (OA), quantity disagreement (QD), and allocation disagreement (AD). PA is the proportion of samples correctly classified as category I in all samples of type I (one column of the confusion matrix). UA indicates, in all samples classified as class I (a row of the confusion matrix), the proportion of samples whose measured type is class I.

OA is the proportion of samples that are correctly classified in all samples. QD and AD are precision evaluation indexes proposed in 2011 [55]. QD is the difference between reference and comparison maps due to incomplete matching of proportions of the categories. AD is the difference between reference and comparison maps due to non-optimal matching in spatial allocation of categories. Before the accuracy calculation, the observed sample matrix was converted into an estimated unbiased population matrix according to reference [55].

## 3. Results

The land-cover classification result maps of seven experiments have been showed in Figure 5.

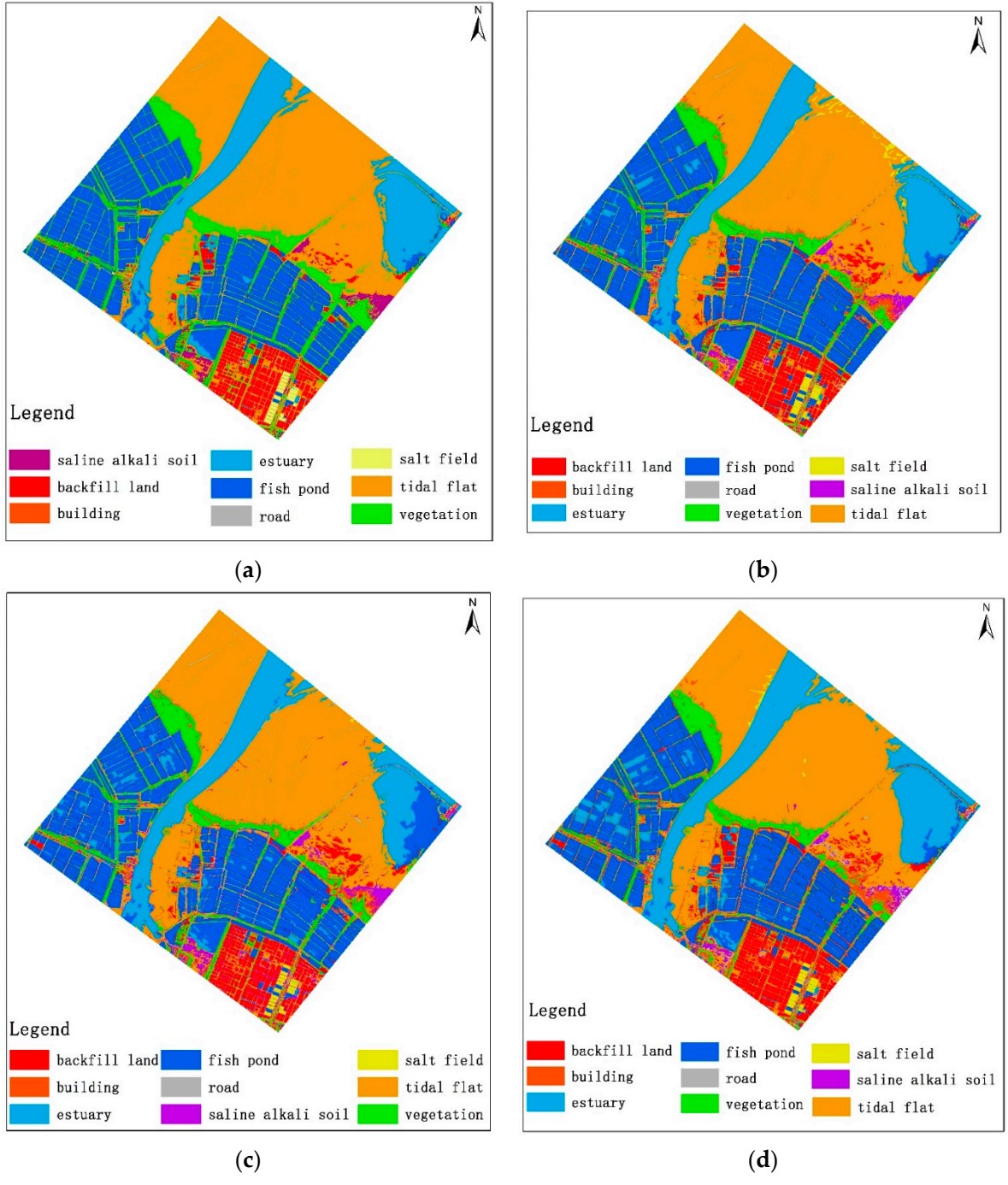

**Figure 5.** *Cont.*

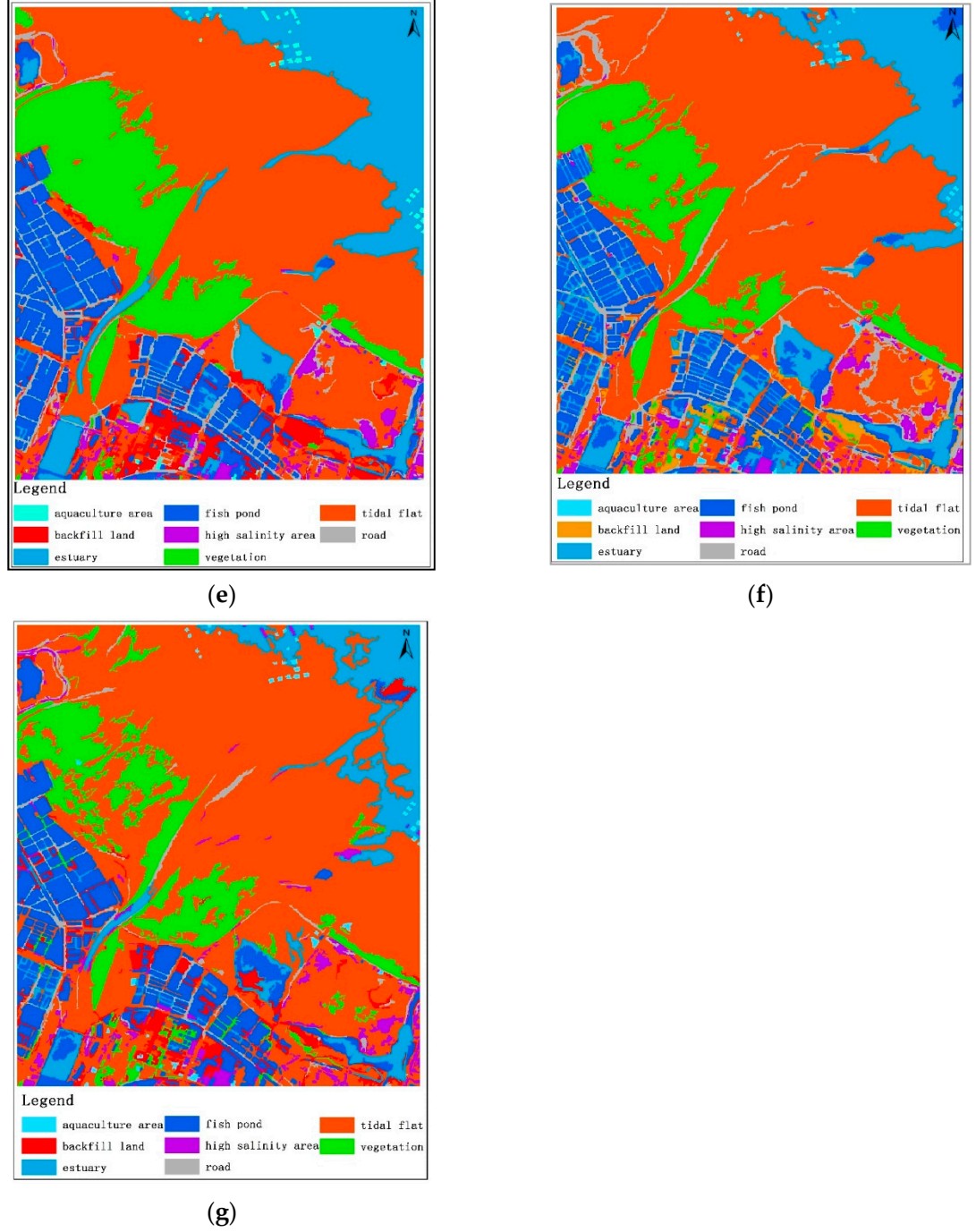

**Figure 5.** The land-cover classification result maps. (**a**) Random forest (RF) method and texture features used in the WV-2 image; (**b**) RF method used in the WV-2 image; (**c**) support vector machine (SVM) method used in the WV-2 image; (**d**) *k*-nearest neighbor (k-NN) method used in the WV-2 image; (**e**) RF method used in the Landsat-8 image; (**f**) SVM method used in the Landsat-8 image; and (**g**) k-NN method used in the Landsat-8 image.

Experiments clearly revealed that in the Worldview-2 images, the overall classification accuracies of RF, SVM, and k-NN were 91.86%, 87.18%, and 87.14%, with QD of 4.61%, 6.06%, and 6.89%, respectively. AD of RF, SVM, and k-NN were 3.53%, 6.76%, and 5.97%. Classification accuracy obtained by adding texture features to the RF algorithm was 94.14%, with a QD of 3.52% and 2.34%.

For Landsat-8 imagery, the overall classification accuracies of RF, SVM, and k-NN were 86.61%, 79.96%, and 77.23%. QD was 6.48%, 7.78%, and 8.43%, respectively. AD was 6.91%, 12.26%, and 14.34% respectively. The image classification results are given in Figure 5.

*3.1. Analysis of Classification Results Using Two Types of Spatial Resolution Satellite Images*

Tables 6–9 give the classification accuracy evaluation results obtained using 0.5 m Worldview-2 imagery; Tables 10–12 give the results obtained using Landsat-8 imagery.

Tables 6–9 give the classification results obtained using 0.5 m Worldview-2 imagery without the texture feature. For Worldview-2 imagery, the eight spectral bands, vegetation index, and five geometric features were all used in the k-NN, SVM, and RF classification methods in Tables 7–9. The overall accuracy of the classification results obtained were 91.86%, 87.18%, and 87.14% for the RF, SVM, and k-NN methods, respectively. According to Tables 7–9, which gives the classification accuracy for each type, the four types of vegetation, backfill land, salt field, and tidal flat were classified with a high degree of accuracy using these three methods—not surprisingly, as these four feature types are very obvious, are easy to distinguish, and do not require complex algorithm support. However, although roads can be well extracted using both the RF and the SVM methods, the k-NN algorithm did not distinguish roads or other types well. The spectral characteristics of the road were quite different, and the roads with different features could be selected as samples by visual interpretation. What is more, the classification results obtained using the k-NN algorithm were greatly affected by the samples. For each object to be classified, the distance to the entire feature space of samples must be calculated, the *k*-nearest neighbors obtained, and the type of the sample to be classified determined. The training sample selection was relatively scattered and could not take advantage of the k-NN algorithm. In the coastal wetland, most of the saline–alkali soil was distributed on the edge of the tidal flat and the spectral characteristics were similar. Using the three classification methods, these two types will be a small amount of misclassification. Estuary and fish pond could be classified using RF and SVM without any misclassification of land cover types among the other six types, whereas using the k-NN method, only one sample was misclassified. The accuracy of estuary and fish pond classification in all three classification methods showed the ability to surpass 90%, indicating that the three methods could distinguish between water and non-water types. However, these three methods did show two kinds of mutual mismatch between the estuary and fish pond.

Building classification is very difficult. Complete building boundaries cannot be extracted for buildings that are covered by vegetation, resulting in low classification accuracy. An improved classification method is needed, or increasing the number of classification features.

Based on the analysis, the RF method was most suitable for the complex land cover types and could be used to classify land cover types using large number of image features. Its classification accuracy was higher than seen for the SVM and k-NN methods in the cases of fewer samples and multiple land cover types. The RF classification method exhibited the highest classification accuracy, indicating that its use in estuary wetland classification plays to its strengths.

Tables 10–12 give the results obtained using Landsat-8 imagery. Using the RF algorithm, overall accuracy of the land cover classification was 86.61% in Landsat-8 imagery. For the SVM and k-NN algorithms, overall accuracies were 79.96% and 77.23%, respectively, using Landsat-8 imagery. The QD of RF, SVM, and k-NN method were 6.48%, 7.78%, and 8.43% respectively. The AD of the three methods were 6.91%, 12.26%, and 14.34%.

In the RF classification method, the classification accuracy of tidal flat, vegetation, backfill land, estuary, and aquaculture area was higher than that of the SVM and k-NN algorithms. For the fish pond category, the features on the image were more obvious and the shape was regular. The classification accuracy of the three methods was relatively high, and could reach more than 90%.

Among the three classification methods, high salinity area, vegetation, backfill land, and estuary were all misclassified with tidal flat in varying degrees. The tidal flat area was large and might include categories such as vegetation, and high salinity area. It was difficult to enter this area during field

surveys. When selecting samples by visual interpretation, these categories could not be distinguished and sample selection might be missed. At the same time, the boundary between the tidal flat and estuary was not obvious, which will lead to the misclassification of the two categories. In the RF algorithm, the PA and UA of the aquaculture area could reach 100%, and the SVM and k-NN algorithms misclassified the aquaculture area into fish ponds with the regular shape feature, indicating that all three classification methods could use shape features to classify, but the RF algorithm had the best effect.

Analysis made it obvious that the three classification methods were more accurate when using Worldview-2 imagery than when using Landsat-8 imagery in images having a lower spatial resolution, a single pixel may contain multiple types of land cover, making it difficult to distinguish between different types. Remote sensing images having lower resolution were not suitable for high-precision classification in estuary wetlands. What is more, the Landsat-8 image classification system differed from that for the Worldview-2 images. According to the different classification system, two different sets of training samples and verification samples were selected for the two images. In Landsat-8 images, it was impossible to visually interpret buildings or to accurately distinguish salt fields from saline alkali soil, and as a result these two types were merged into the category of high-salinity areas. High-resolution remote sensing images, by contrast, could show actual distributions and specific details of terrain, improving classification accuracy. Regardless of the difference between WV-2 and Landsat-8 image resolution, the eight land cover types could be expressed in Landsat-8 images with the RF method. The overall accuracy could reach 86.61%, higher than SVM and k-NN. The spectral characteristics of Landsat-8 would affect the classification accuracy. In addition, the boundary between different land cover types was not clear, which would lead to a decrease in classification accuracy. In our study, Landsat-8 images were obtained free and could be used for experiments after pre-processing, which was economical and convenient. The specific size of the image could also be obtained according to research needs.

**Table 6.** Estimated population matrix of the RF method in the WV-2 image with additional texture features.

| | Vegetation | Road | Building | Backfill Land | Salt Field | Saline Alkali Soil | Tidal Flat | Estuary | Fish Pond | Comparison Total |
|---|---|---|---|---|---|---|---|---|---|---|
| Vegetation | 0.1470 | | 0.0090 | | | | | | | 0.1561 |
| Road | | 0.0412 | | | | 0.0034 | | | | 0.0446 |
| Building | | 0.0036 | 0.1075 | | | 0.0036 | | | | 0.1146 |
| Backfill Land | | | | 0.0669 | | | | | | 0.0669 |
| Salt Field | | | | | 0.0255 | | | 0.0032 | | 0.0287 |
| Saline Alkali Soil | | | | | | 0.0510 | | | | 0.0510 |
| Tidal Flat | | | 0.0087 | | 0.0029 | 0.0058 | 0.1932 | | | 0.2134 |
| Estuary | | | | | | | | 0.1375 | 0.0122 | 0.1497 |
| Fish Pond | | | | | | | | 0.0034 | 0.1718 | 0.1752 |
| Reference Total | 0.1470 | 0.0476 | 0.1251 | 0.0669 | 0.0284 | 0.0637 | 0.1932 | 0.1440 | 0.1840 | 1 |
| PA | 1 | 0.8822 | 0.8593 | 1 | 0.8979 | 0.08006 | 1 | 0.9549 | 0.9337 | |
| UA | 0.9417 | 0.9238 | 0.9380 | 1 | 0.8885 | 1 | 0.9053 | 0.9185 | 0.9806 | |

Quantity disagreement (QD) = 0.0352. Allocation disagreement (AD) = 0.0234. Overall accuracy (OA) = 0.9414.

**Table 7.** Estimated population matrix of the RF method in the WV-2 image.

| | Vegetation | Road | Building | Backfill Land | Salt Field | Saline Alkali Soil | Tidal Flat | Estuary | Fish Pond | Comparison Total |
|---|---|---|---|---|---|---|---|---|---|---|
| Vegetation | 0.1416 | | 0.0144 | | | | | | | 0.1561 |
| Road | | 0.0409 | | | | 0.0037 | | | | 0.0446 |
| Building | | | 0.1146 | | | | | | | 0.1146 |
| Backfill Land | | | 0.0029 | 0.0611 | | | 0.0029 | | | 0.0669 |
| Salt Field | | | | | 0.0287 | | | | | 0.0287 |
| Saline Alkali Soil | | 0.0051 | 0.0127 | | | 0.0331 | | | | 0.0510 |
| Tidal Flat | | 0.0029 | 0.0088 | | 0.0029 | 0.0058 | 0.1929 | | | 0.2134 |
| Estuary | | | | | | | | 0.1401 | 0.0096 | 0.1497 |
| Fish Pond | | | | | | | | 0.0096 | 0.1656 | 0.1752 |
| Reference Total | 0.1416 | 0.0489 | 0.1535 | 0.0611 | 0.0316 | 0.0427 | 0.1958 | 0.1497 | 0.1752 | 1 |
| PA | 1 | 0.8364 | 0.7466 | 1 | 0.9082 | 0.7752 | 0.9852 | 0.9359 | 0.9452 | |
| UA | 0.9071 | 0.9170 | 1 | 0.9133 | 1 | 0.6490 | 0.9039 | 0.9359 | 0.9452 | |

QD = 0.0461. AD = 0.0353. OA = 0.9186.

**Table 8.** Estimated population matrix of the SVM method in the WV-2 image.

| | Vegetation | Road | Building | Backfill Land | Salt Field | Saline Alkali Soil | Tidal Flat | Estuary | Fish Pond | Comparison Total |
|---|---|---|---|---|---|---|---|---|---|---|
| Vegetation | 0.1416 | | 0.0116 | 0.0029 | | | | | | 0.1561 |
| Road | | 0.0446 | | | | | | | | 0.0446 |
| Building | | 0.0156 | 0.0938 | | | 0.0052 | | | | 0.1146 |
| Backfill Land | | | 0.0035 | 0.0634 | | | | | | 0.0669 |
| Salt Field | | | | | 0.0287 | | | | | 0.0287 |
| Saline Alkali Soil | | | 0.0096 | | | 0.0414 | | | | 0.0510 |
| Tidal Flat | | | 0.0260 | 0.0052 | 0.0026 | 0.0052 | 0.1743 | | | 0.2134 |
| Estuary | | | | | | | | 0.1257 | 0.0239 | 0.1497 |
| Fish Pond | | | | | | | | 0.0168 | 0.1583 | 0.1752 |
| Reference Total | 0.1416 | 0.0602 | 0.1445 | 0.0715 | 0.0313 | 0.0518 | 0.1743 | 0.1426 | 0.1823 | 1 |
| PA | 1 | 0.7409 | 0.6491 | 0.8867 | 0.9169 | 0.7992 | 1 | 0.8815 | 0.8683 | |
| UA | 0.9071 | 1 | 0.8185 | 0.9477 | 1 | 0.8118 | 0.8168 | 0.8397 | 0.9035 | |

QD = 0.0606. AD = 0.0676. OA = 0.8718.

**Table 9.** Estimated population matrix of the k-NN method in the WV-2 image.

| | Vegetation | Road | Building | Backfill Land | Salt Field | Saline Alkali Soil | Tidal Flat | Estuary | Fish Pond | Comparison Total |
|---|---|---|---|---|---|---|---|---|---|---|
| Vegetation | 0.1390 | | 0.0170 | | | | | | | 0.1561 |
| Road | | 0.0324 | 0.0081 | | | 0.0041 | | | | 0.0446 |
| Building | | 0.0060 | 0.1086 | | | | | | | 0.1146 |
| Backfill Land | | | | 0.0635 | | | | | 0.0033 | 0.0669 |
| Salt Field | | | | | 0.0287 | | | | | 0.0287 |
| Saline Alkali Soil | | 0.0089 | 0.0133 | | | 0.0288 | | | | 0.0510 |
| Tidal Flat | | 0.0028 | 0.0111 | 0.0055 | 0.0028 | 0.0055 | 0.1857 | | | 0.2134 |
| Estuary | | | | | | | | 0.1238 | 0.0259 | 0.1497 |
| Fish Pond | | | | | | | | 0.0143 | 0.1609 | 0.1752 |
| Reference Total | 0.1390 | 0.0501 | 0.1581 | 0.0691 | 0.0314 | 0.0384 | 0.1857 | 0.1381 | 0.1901 | 1 |
| PA | 1 | 0.6467 | 0.6869 | 0.9190 | 0.9140 | 0.7500 | 1 | 0.8965 | 0.8464 | |
| UA | 0.8905 | 0.7265 | 0.9476 | 0.9492 | 1 | 0.5647 | 0.8702 | 0.8270 | 0.9184 | |

QD = 0.0689. AD = 0.0597. OA = 0.8714.

**Table 10.** Estimated population matrix of the RF method in the Landsat-8 image.

|  | Tidal Flat | High Salinity Area | Vegetation | Backfill Land | Fish Pond | Estuary | Road | Aquaculture Area | Comparison Total |
|---|---|---|---|---|---|---|---|---|---|
| Tidal Flat | 0.1183 | 0.0197 | 0.0039 | 0.0079 |  | 0.0158 | 0.0237 |  | 0.1893 |
| High Salinity Area | 0.0148 | 0.0962 |  |  |  |  | 0.0074 |  | 0.1183 |
| Vegetation |  |  | 0.0888 |  |  |  |  |  | 0.0888 |
| Backfill Land |  |  |  | 0.0592 |  |  |  |  | 0.0592 |
| Fish Pond |  |  |  | 0.0057 | 0.2434 | 0.0113 |  |  | 0.2604 |
| Estuary |  |  |  |  | 0.0084 | 0.0922 |  |  | 0.1006 |
| Road |  | 0.0153 |  |  |  |  | 0.1149 |  | 0.1302 |
| Aquaculture Area |  |  |  |  |  |  |  | 0.0533 | 0.0533 |
| Reference Total | 0.1331 | 0.1312 | 0.0927 | 0.0727 | 0.2518 | 0.1193 | 0.1459 | 0.0533 | 1 |
| PA | 0.8888 | 0.7332 | 0.9579 | 0.8143 | 0.9666 | 0.7728 | 0.7875 | 1 |  |
| UA | 0.6249 | 0.8132 | 1 | 1 | 0.9347 | 0.9165 | 0.8825 | 1 |  |

QD = 0.0648. AD = 0.0691. OA = 0.8661.

**Table 11.** Estimated population matrix of the SVM method in the Landsat-8 image.

|  | Tidal Flat | High Salinity Area | Vegetation | Backfill Land | Fish Pond | Estuary | Road | Aquaculture Area | Comparison Total |
|---|---|---|---|---|---|---|---|---|---|
| Tidal Flat | 0.1092 | 0.0146 | 0.0109 | 0.0109 |  | 0.0291 | 0.0146 |  | 0.1893 |
| High Salinity Area | 0.0139 | 0.0975 |  |  |  |  | 0.0070 |  | 0.1183 |
| Vegetation |  |  | 0.0666 | 0.0222 |  |  |  |  | 0.0888 |
| Backfill Land |  |  |  | 0.0296 |  |  |  | 0.0296 | 0.0592 |
| Fish Pond |  |  |  |  | 0.2488 | 0.0116 |  |  | 0.2604 |
| Estuary |  |  |  | 0.0112 | 0.0112 | 0.0782 |  |  | 0.1006 |
| Road |  | 0.0137 |  |  |  |  | 0.1165 |  | 0.1302 |
| Aquaculture Area |  |  |  |  |  |  |  | 0.0533 | 0.0533 |
| Reference Total | 0.1232 | 0.1257 | 0.0775 | 0.0739 | 0.2600 | 0.1189 | 0.1380 | 0.0828 | 1 |
| PA | 0.8864 | 0.7757 | 0.8594 | 0.4005 | 0.9569 | 0.6577 | 0.8442 | 0.6437 |  |
| UA | 0.5769 | 0.8242 | 0.7500 | 0.5000 | 0.9555 | 0.7773 | 0.8948 | 1 |  |

QD = 0.0778. AD = 0.1226. OA = 0.7996.

**Table 12.** Estimated population matrix of the k-NN method in the Landsat-8 image.

| | Tidal Flat | High Salinity Area | Vegetation | Backfill Land | Fish Pond | Estuary | Road | Aquaculture Area | Comparison Total |
|---|---|---|---|---|---|---|---|---|---|
| Tidal Flat | 0.1114 | 0.0223 | | 0.0037 | | 0.0223 | 0.0297 | | 0.1893 |
| High Salinity Area | | 0.0947 | | | | 0.0079 | 0.0158 | | 0.1183 |
| Vegetation | | | 0.0740 | 0.0148 | | | | | 0.0888 |
| Backfill Land | | | | 0.0444 | | | | 0.0148 | 0.0592 |
| Fish Pond | | | | 0.0055 | 0.2327 | 0.0111 | | 0.0111 | 0.2604 |
| Estuary | 0.0091 | | | | | 0.0183 | 0.0732 | | 0.1006 |
| Road | 0.0087 | 0.0174 | | 0.0087 | | | 0.0955 | | 0.1302 |
| Aquaculture Area | | | | 0.0067 | | | | 0.0466 | 0.0533 |
| Reference Total | 0.1292 | 0.1343 | 0.0740 | 0.0838 | 0.2509 | 0.1144 | 0.1557 | 0.0577 | 1 |
| PA | 0.8622 | 0.7051 | 1 | 0.5298 | 0.9275 | 0.6399 | 0.6134 | 0.8076 | |
| UA | 0.5885 | 0.8005 | 0.8333 | 0.7500 | 0.8936 | 0.7276 | 0.7335 | 0.8743 | |

QD = 0.0843. AD = 0.1434. OA = 0.7723.

*3.2. Analysis of Classification Results When Fewer Training Samples are Available*

When classifying wetlands, sometimes only a small number of samples of certain types are obtained after image scale segmentation. For example, using Worldview-2 imagery, only 14 training samples of the road class were available. When a classification training sample is small, classification accuracy decreases. For road classification, using the RF algorithm, PA and UA were 83.64% and 91.70%, respectively. The SVM algorithm produced a similar classification result, but the k-NN algorithm gave a PA of only 74.09%. Using Landsat-8 imagery, backfill land only had 10 training samples, so that classification PA for this land class were 81.43%, 40.05%, and 52.98% for the RF, SVM, and k-NN algorithms, with a UA of 100%, 50.00%, and 75.00%, respectively.

From these results, it can be seen that, using 0.5 m Worldview-2 imagery, both the RF and the SVM algorithms achieved better classification results based on scaled objects when few samples were available. Using Landsat-8 imagery, a number of backfill lands could be omitted when using the SVM and k-NN algorithms to classify the image. When using the RF algorithm, fewer objects were omitted or mixed than when using other methods. These results coincide with those of extant studies and indicate one advantage of the RF method: When the number of samples is small, higher classification accuracy can be obtained [56].

*3.3. Texture Features' Influence on Classification Accuracy*

The RF algorithm has distinct advantages when using a large number of features for classification. As a result, texture features were added to the classification features and the influence of texture features on RF algorithms' classification accuracy studied. In the RF classification method, in addition to spectral features, exponential features, and geometric features, texture features were added for the classification of Worldview-2 imagery, obtaining an improved overall classification accuracy of 94.14%, QD of 3.52%, and AD of 2.34%. Vegetation, backfill land, and salt field were classified by a different combinations of features, and the confusion matrix was identical, indicating that these three types of land cover were clearly characterized by a certain spectrum, index, and geometry features that could lead to highly accurate classification. Adding texture features could improve classification accuracy for the road, saline alkali soil, tidal flat, estuary, and fish pond types, to a degree, and could greatly improve the classification accuracy of buildings while leading to fewer misclassifications between buildings and both backfill land and saline alkali soil. As a result, use of texture features could improve accuracy in the classification of estuary wetlands.

## 4. Discussion

When using the same classification method and the same features, the high-resolution Worldview-2 could achieve higher classification accuracy than the Landsat-8 image in the wetland land cover study in small areas. At the same time, Worldview-2 images could express the land cover type in more detail. Through visual interpretation, Worldview-2 images could represent nine different land cover types, while Landsat-8 images could only display eight types. This results in different classification systems and classification samples for the two images. In addition to describing the actual resolution of the image, the definition of "high resolution" needs to be based on the range of the study area. In our small-range study area, the Landsat-8 image with 15 m spectral resolution had a certain difference in the accuracy of the classification results from the Worldview-2 image with a spectral resolution of 0.5 m. However, in the global forest cover change study conducted by Hansen et al. [57], the 30 m Landsat image is a good choice. At the same time, due to the earlier and continuous Landsat satellite launch, long-term sequence studies can be performed. In addition, global Landsat images are available for free.

Our experiments compared the adaptability of three algorithms to land cover classification in small-range coastal wetlands. The selection of samples and features were relatively simple and easy to implement. Compared with the research of Hansen et al. [57] and Peekel et al. [58], the amount of data processing and corresponding reference materials were less. Hansen et al. and Peekel et al.

respectively studied global forest cover change and global surface water change. These two studies are essentially to study the distribution of a land cover type. Our experiment classified all land cover types in the image.

In the aspect of sample selection, our experiment was based on field investigation combined with visual interpretation. Different training samples and validation samples were selected for two different resolution images. This sample selection method is feasible when the research area is small and the amount of data is small. In the study of Gong Peng et al. [59], a stable classification method using limited samples has been proposed to use the Landsat sample set with global resolution of 30 m for global land cover mapping with 10 m resolution. This sample transformation method enables the sample data set to play its maximum value. Sample transformation reduces the time of sample selection when doing large-area research. In different scales of research, it can ensure the uniformity of samples, and the experimental results are more reliable. Similarly, the RF method is also used in land cover mapping by Gong Peng et al. Therefore, the RF method can be applied not only to large-scale land use classification of coastal wetlands, but also to global land use classification.

In the future, there are several aspects that can be optimized and further studied throughout the classification process. In the object-oriented segmentation process, how to set the optimal segmentation scale and optimally express the land cover type. In terms of feature selection, information such as location features can be added to improve the accuracy of the classification. In classification algorithms, in addition to the RF method, other deep learning algorithms such as deep convolutional neural networks can be used.

## 5. Conclusions

The high-resolution remote sensing image could describe the actual object in detail and reflect the real situation of the ground object. The high-resolution Worldview-2 image was more suitable for the small-scale land cover type classification than the medium-resolution Landsat-8 image. The classification system of Worldview-2 images was more detailed than Landsat-8 images, and the classification accuracy was higher. Landsat-8 image classification accuracy was low due to spatial resolution. Landsat-8 image was easy to obtain and had a wide coverage, which was suitable for the large-scale land cover studies. The RF algorithm demonstrated its superiority in handling a large number of features. For Worldview-2 and Landsat-8 images, the use of RF algorithms for classification of almost all land cover types was more accurate than with SVM and k-NN algorithms. At the same time, the RF algorithm provided a more accurate classification for a small number of samples, and was more suitable for estuarine wetland classification than the SVM or k-NN algorithm.

In addition, different texture features described the local pattern and arrangement rules of image from different aspects. The classification accuracy when using the RF algorithm could be improved by adding texture features.

In general, high-resolution remote sensing images were suitable experimental data for small-scale land cover classification of coastal wetlands. The RF algorithm could give full play to its advantages and obtain high-quality classification results. What is more, among many classification features, the texture features were one of the classification features, which could improve the accuracy of classification.

**Author Contributions:** Investigation, X.W., Z.C., J.W., Y.Z., X.L. and H.Z.; Methodology, X.W., X.F., Z.C., J.W. and Y.Z.; Writing—Original draft, X.W. and X.F.; Writing—Review & editing, X.G., Y.Z. and Y.F.; Improving the data analysis, Y.Z. and X.F.

**Funding:** This research was funded by the National Key Research and Development Program of China (Project Ref. No. 2016YFB0501501), the Natural Science Foundation of China (NSFC No. 31270745; No. 41506106), Lianyungang Land and Resources Project (LYGCHKY201701), Lianyungang Science and Technology Bureau Project (SH1629), the Priority Academic Program Development of Jiangsu Higher Education Institutions (PAPD), the Top-notch Academic Programs Project of Jiangsu Higher Education Institutions (TAPP), and Jiangsu Postgraduate Innovation Program (No. 5508201601; No. SY201808X).

**Acknowledgments:** Worldview-2 satellite imagery data is highly appreciated.

**Conflicts of Interest:** The authors declare no conflict of interest.

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
