# Peer review of "Land-Cover Classification of Coastal Wetlands Using the RF Algorithm for Worldview-2 and Landsat 8 Images"

_remotesensing, doi:10.3390/rs11161927_

Round 1
Reviewer 1 Report
The theme of the manuscript is relevant and scientifically valid. On the other hand, the manuscript is misleading, very confusing, poorly structured and requires profound changes to be able to publish in the Remote Sensing journal. Here are my considerations:
1) Conceptual Aspects:
1.1) The comparison between Worldview-2 and Landsat-8 needs to be more fair. I have no doubt that Worldview-2 is fantastic, and is more suitable for small-scale land cover classification (as indicated on line 387) and it will produce better classification results. On the other hand, it is private data and does not have global coverage like Landsat-8. The manuscript should clearly tell to the reader in which situations it is best to use Wordlview-2 or Landsat-8, indicating real tradeoffs.
1.2) Your classification approach is interesting and produced good results, however it is not operation or applicable for huge regions, as all coastal wetland of China. The most part of nowadays classification approaches are combining cloud-computing, machine learning algorithms and huge volume of data (see the references), and you don't discuss it. The manuscript needs discuss it, comparing and pointing the real advantages and disadvantages of your classification approach.
1.3) It is really frustrating don't see any classification result/map in your manuscript. You ran 8 classifications and you did not bother to show a result even. This is a remote sensing journal you need show and analysis the classification results. Only accuracy metrics are not enough.
2) Manuscript Structure:
2.1) The manuscript structure is very confusing. Please, considering use something like this:
1. Introduction
2. Data and Methods:
2.1 Study Area
2.2 Satellite Data
2.3 Training and Verification Data
2.4 Segmentation
2.5 Features
2.6 Accuracy evaluation
3. Results
4. Discussion
6. Conclusion
2.2) You need more references in the first paragraph, that supports what you writed in the lines 51-58
2.3) Lines 66-70 are wrong, because nowadays we have Sentinel-1 as public available radar data. By the way all this discussion of radar data is superfluous, and it is not related with the manuscript theme and doesn't aggregate anything to text. My suggestion is remove.
2.4) One sentence is not a paragraph. (Line 101)
2.5) The last paragraph of introduction could be better as the introduction of section "2. Data and Methods"
2.6) In table 2 there are several classes without images.
2.7) You need describe better the classification tests using texture features, it is very confusing in the text.
3) Technical Approaches:
3.1) Where is your stratification approach and your strategy to collect the training and verification data ? It is not clear in the manuscript how you implement that. Do you collect points or polygons ? Do you classify them using field work or visual interpretation ?
3.2) Do you balanced your confusion matrixes according with the mapped areas ? It's a really important step to produce a unbiased estimates. Don't use kappa (see Pontius & Millones, 2011).
3.3) What parameters do you use for k-NN, SVM, Random Forest ? It's really important inform them.
3.4) Specifically on SVM, a really important parameter of the algorithm is the kernel function. The choosing of kernel can significantly alter the performance of the classifier, producing very good results. What kernel function did you choose and why ? It's really important because can change your conclusions.
Classification Approaches References:
Hansen, M.C., Potapov, P.V., Moore, R., Hancher, M., Turubanova, S.A., Tyukavina, A., Thau, D., Stehman, S.V., Goetz, S.J., Loveland, T.R., Kommareddy, A., Egorov, A., Chini, L., Justice, C.O., Townshend, J.R.G., 2013. High-resolution global maps of 21st-century forest cover change. Science 342 (6160), 850–853.
Pekel, J.F., Cottam, A., Gorelick, N., Belward, A.S., 2016. High-resolution mapping of global surface water and its long-term changes. Nature 540 (7633), 418.
Gong, P., Liu, H., Zhang, M., Li, C., Wang, J., Huang, H., ... & Chen, B. Stable classification with limited sample: transferring a 30-m resolution sample set collected in 2015 to mapping 10-m resolution global land cover in 2017. Science Bulletin, v. 64, n. 6, p. 370-373, 2019.
Author Response
Dear Reviewer1,
Thanks for your comments and suggestions.
We have revised and updated as attached with responses to all comments.
Yours sincerely,
Authors

Reviewer 2 Report
generally well written, easy to read and follows, no significant typos in the text, apart from a few slighly convoluted sentences (see for instance line 86-87). well documented and abstract, l22-23: clear advantages relative to what other algorithms-approaches? pls explain-expand. same in L28, advantages of the RF algorithm over what? this is given in the body (lines121-130, for instance, but should probably be cited in the abstract too) L31: confidence levels and/or significance levels for correlation? l32: are the 4-5% improvement significant in terms of results, accuracy, computation costs? l35-37: quantify the improvement when coverage or data quality is poor Introduction: line 50-52: not only land use but also due to climate changes-sea-level increase and similar L98: check citation for consistency with journal policy L103: what does 'theoretically mature' mean? L103-105 rephrase, not clear L161-162: not clear- the estuary is 24.7km2, while later on a 'thousands of square km' is used instead. I reckon this is the catchment area of the rivers listed here, not the estuary. pls.clarify Figure 1: check label as 'F' is missing Section 2-3-4: check font size Figure 2: please redraw it as it is poor quality and add verbosity to the caption; I know all is described properly in the ms but it should be described more throuughly please add a 'Discussion' Section as wellAuthor Response
Dear Reviewer2,
Thanks for your comments and suggestions.
We have revised and updated as attached with responses to all comments.
Yours sincerely,
Authors

Reviewer 3 Report
Line 42-43: end sentence after ... k-NN algorithms are. The next 2.5 lines are just a repetition of what was said before --> delete
Literature quotation requires major revision: e.g.
Line 95: Sader et al. is missing
Line 117: Salehi.B. is referenced as Mahdavi, Sahel, Bahram Salehi (Line 477). I would recommend to harmonize all across the text (e.g. cite "M.S.B. Salehi" or only "Sahlehi) and add either the date or directly behind the name the reference number
Line 130: Mahdianpari, M., et al is missing (not reference 32)
General remark on citations: in the Reference section (Line 406 ff.): first names are sometimes abbreviated sometimes fully written --> please harmonize!
Line 198 f.: please describe better how field survey was carried out and how results were stored (e.g. GNSS measured points, tables, photographs etc.)
Line 208 f. (Table 2): it would be good to provide information about the size of the samples (e.g. in square meters) to allow a better understanding of the representativity of the samples.
Line 210: Figure 2: what meaning do have the various colours in c and d?
Line 261 f.: It would support the understanding of your results if you could provide the confusion (error) matrix created together with its analysis in Table 5.
Line 334: Chapter 5.3: As you already describe the differences between Worldview & Landsat in Chapter 5.1 this chapter should come either before or should be merged into 5.1
Line 369 f.: Could you say something about the additional effort using extensive texture features? This would allow to estimate the suitability of your method for transferring it into operational wetland assessments in other regions.
Line 384 f.: What will be the next steps? Will you use the approach now to analyse wetlands as a common approach (i.e. operationalize it)?
Author Response
Dear Reviewer3,
Thanks for your comments and suggestions.
We have revised and updated as attached with responses to all comments.
Yours sincerely,
Authors

Round 2
Reviewer 1 Report
2) Manuscript Structure:
2.1) The last paragraph of the 1. Introduction is absolutely the same as the first paragraph of the 2. Data and Method. You need to close the introduction with a paragraph that gives a comprehensive overview of the manuscript.
2.2) Please, revise the figure legends and use the correct standard (without description - lines 223-225).
2.3) Your conclusion has less than 10 lines. It is really unusual. Could you improve it, pointing the key aspects of the manuscript and same possible next steps ?
3) Technical Approaches:
3.1) Your training and verification data was polygons. How they were delimited ? By hand draw or by a automatic segmentation approach ? It needs be more clear in the manuscript.
Author Response
Dear Reviewer,
Thanks for your comments. We have revised and updated as suggested.
Yours sincerely,
Authors

Reviewer 3 Report
thank you for the complete rework of the paper. It is now much better.
Author Response

(The authors gave the same response as above.)
